# Systematic Investigation of the Effects of Multiple SV40 Nuclear Localization Signal Fusion on the Genome Editing Activity of Purified SpCas9

**DOI:** 10.3390/bioengineering9020083

**Published:** 2022-02-21

**Authors:** Sailan Shui, Shaojie Wang, Jia Liu

**Affiliations:** Shanghai Institute for Advanced Immunochemical Studies, School of Life Science and Technology, ShanghaiTech University, Shanghai 201210, China; shuisl2016@gmail.com (S.S.); wangshj@shanghaitech.edu.cn (S.W.)

**Keywords:** CRISPR-Cas9, Cas9 ribonucleoproteins (RNPs), nuclear localization signal (NLS), genome editing

## Abstract

The emergence of CRISPR-Cas9 technology has revolutionized both basic and translational biomedical research. For Cas9 nuclease to exert genome editing activity, nuclear localization signal (NLS) derived from simian virus 40 (SV40) T antigen is commonly installed as genetic fusion to direct the intracellular Cas9 proteins to the nucleus of cells. Notably, previous studies have shown that multiple SV40 NLS fusion can improve the targeting activity of Cas9-derived genome-editing and base-editing tools. In addition, the multi-NLS fusion can increase the intracellular activity of Cas9 in the forms of both constitutive expression and directly delivered Cas9-guide RNA ribonucleoprotein (RNP) complex. However, the relationship between NLS fusion and intracellular Cas9 activity has not been fully understood, including the dependency of activity on the number or organization of NLS fusion. In the present study, we constructed and purified a set of Streptococcus pyogenes Cas9 (SpCas9) variants containing one to four NLS repeats at the N- or C-terminus of the proteins and systematically analyzed the effects of multi-NLS fusion on the activity of SpCas9 RNPs. It was found that multi-NLS fusion could improve the intracellular activity as lipofected or nucleofected Cas9 RNPs. Importantly, multi-NLS fusion could enhance the genome-editing activity of SpCas9 RNPs in primary and stem/progenitor cells and mouse embryos.

## 1. Introduction

Clustered regularly interspaced short palindromic repeat (CRISPR)-CRISPR-associated proteins (Cas) is the bacterial adaptive immune system for defending bacteriophage infection [1,2]. CRISPR-Cas9 contains a single nuclease domain and can be engineered to function with a single guide RNA (sgRNA) [3]. This modular feature has made CRISPR-Cas9 system feasible for versatile genome editing applications [4,5,6,7]. Importantly, Cas nucleases including *Streptococcus pyogenes* Cas9 (SpCas9) can be genetically fused with different effector domains to mediate base editing, prime editing and transposition [8]. Despite the myriad forms of SpCas9-derived fusion proteins, they all contain one or more simian virus 40 (SV40) large T antigen nuclear localization signals (NLS) for the nuclear transport of intracellular SpCas9 proteins.

SV40 NLS with the amino acid sequence PKKKRKV [9] is commonly used in the ectopic expression of genome-editing tools including zinc finger proteins (ZFPs), transcription activator-like effectors (TALEs) and Cas proteins [10] to exert their activity on genomic DNA. In conventional set up, these genome-editing tools are fused with one SV40 NLS sequence. However, additional SV40 NLS fusion has been found to be capable of increasing the intracellular activity of directly delivered ZFN proteins [11]. Likewise, extra SV40 NLS can improve the efficiency of SpCas9-derived genome-editing tools including nucleases [12], transcription activators [13] and base editors [14]. Importantly, the advantageous effects of multiple NLS repeats appeared to be independent of the forms of SpCas9 cargos. The cellular activity of SpCas9 that is delivered as plasmids [13], mRNA [12] or proteins [15] can be all improved by multi-NLS fusion.

Due to the notable beneficial effects, multi-NLS organization has been used in a variety of genome engineering applications as genetic fusion to SpCas9. Multi-NLS fusion has been found to be capable of improving the nuclear localization of catalytically inactive Cas9 (dCas9) in fusion with fluorescent proteins for enhanced labelling and visualization of genomic loci [16]. Multi-NLS has been employed under therapeutic settings to improve the cellular activity of SpCas9 RNPs in difficult-to-transfect hematopoietic stem cells (HSCs) [15]. SpCas9 RNPs carrying multi-NLS fusion can be combined with black phosphorus nanosheets to mediate efficient genome editing [17]. More interestingly, SpCas9 proteins harboring multi-NLS fusion exhibit enhanced activity when directly delivered into mouse brains [18] or Zebrafish [19]. Moreover, multi-NLS fusion can improve the genome-editing activity of CRISPR-Cas12a in mammalian cells [20]. A recent study also showed that multi-NLS architecture could be combined with adenoviral vector and enhance the gene therapy efficiency of Cas9 [21].

Despite the widespread application of multi-NLS for CRISPR-Cas-mediated genome engineering applications, the effects of multi-NLS on CRISPR-Cas have not been systematically investigated. In the present study, we focused on purified SpCas9 proteins and examined the effects of multi-NLS fusion on the cellular activity of SpCas9 RNPs. Importantly, we assessed the effects of multi-NLS fusion in the contexts of different delivery approaches including lipofection, nucleofection and microinjection and of different cell types such as transformed cell lines and primary or stem/progenitor cells (Figure 1). Particularly, we found that fusion of up to four NLS repeats to the C-terminus of SpCas9 could improve the genome-editing activity of SpCas9 RNPs. Therefore, we study represents the first comprehensive analysis of multi-NLS SpCas9 and paves the way to RNP-based genome editing using multi-NLS SpCas9.

## 2. Materials and Methods

### 2.1. Expression and Purification of Cas9 Proteins

The Cas9 proteins described in the present study were expressed and purified using the same procedure as described below. pET-28b plasmids coding Cas9 proteins were transformed into BL21 (DE3) cells. Single colonies were grown in 800 mL Luria-Bertani (LB) medium supplemented with 50 μg/mL kanamycin. Culture was grown to an OD_600_ of 0.8 and protein expression was induced with 0.2 mM isopropyl-β-d-thiogalactopyranoside (IPTG) overnight at 16 °C. Cells from 800 mL culture were pelleted by centrifugation at 5000× *g* for 10 min and then re-suspended in 40 mL binding buffer containing 20 mM Tris-HCl, pH 8.0, 0.5 M NaCl, 1 mM Tris (2-carboxyethyl) phosphine (TCEP) and 1× complete inhibitor cocktail (Roche). Cells were lysed by sonication on ice, followed by centrifugation at 25,000× *g* at 4 °C for 30 min. The supernatant of cell lysate was incubated with 1 mL Ni-NTA agarose beads (QIAGEN) at 4 °C for 1 h. The protein-bound resin was then washed with 20 mL wash buffer containing 20 mM Tris-HCl, pH 8.0, 0.5 M NaCl and 50 mM imidazole. Proteins were eluted with 5 mL elution buffer containing 20 mM Tris-HCl, pH 8.0, 0.5 M NaCl and 300 mM imidazole. The eluted proteins were buffer exchanged to storage buffer containing 20 mM HEPES, pH 8.0 and 200 mM NaCl, aliquoted and stored at −80 °C.

### 2.2. In Vitro Transcription of sgRNA

sgRNA was transcribed from a sgRNA-coding PCR product with a 5′ T7 promoter sequence using HiScribe T7 Quick High yield RNA Synthesis kit (NEB) (Appendix A). The transcription reaction was performed at 37 °C overnight and then purified by phenol: chloroform extraction, followed by ethanol precipitation. Purified sgRNA was quantified by spectrometry and stored at −80 °C.

### 2.3. In Vitro Cleavage Assay

Cas9 protein and transcribed *CCR5*-targeted sgRNA were mixed and diluted with reaction buffer containing 1× NEB 3.1 buffer supplemented with 1 mM DTT. Cleavage was performed in 10 μL reactions containing 100 ng of substrate DNA and 1 μL RNP complex at indicated concentrations at room temperature for 1 h. Reactions were terminated by addition of 1× DNA loading buffer and resolved on 1% agarose gels.

### 2.4. Cell Lines, Primary Cells and Stem Cells

All cells were cultured at 37 °C in a humidified atmosphere with 5% CO_2_. K562 and Jurkat cells were acquired from American Type Culture Collection (ATCC) and maintained in RPMI 1640 medium containing 10% (*v*/*v*) fetal bovine serum (FBS, Gibco), 100 U/mL penicillin and 100 U/mL streptomycin. Hepa 1-6 and HEK293-derived EGFP reporter cells were maintained in Dulbecco’s modified Eagle’s medium (DMEM) supplemented with 10% (*v*/*v*) FBS, 100 U/mL penicillin and 100 U/mL streptomycin. The medium of EGFP reporter contained an additional 50 U/mL hygromycin. Human dermal fibroblasts (hDF, AllCells) were acquired from Cell Applications (Lot No. 116-500) and maintained in human dermal fibroblast medium. Neural progenitor cells (NPCs) were acquired from American Type Culture Collection(ATCC; ATCC-BXS0117; ACS-5003^TM^) and maintained in NPC complete growth medium (Lonza, Basel, Switzerland, CC-3209) in plates pre-treated with 5% Matrigel (BD Bioscience, Franklin Lakes, NJ, USA, 354234). All the cells used in this study were mycoplasma free, examined using PCR (Appendix A).

### 2.5. Nucleofection

RNP nucleofection was performed with 2 × 10^5^ K562 or Jurkat cells using Lonza 4D nucleofector (Lonza) following the machine’s pre-set K562 and Jurkat transfection programs [22]. Briefly, Cas9 proteins and sgRNA were pre-assembled at indicated concentrations at room temperature for 10 min, mixed with suspended cells and then subjected to electroporation according to manufacturer’s instructions. Unless noted otherwise, cells were harvested at 48 h post nucleofection and subjected to T7E1 assay using T7E1 endonuclease (NEB) as described [22]. For sequence analysis, CRISPR-Cas9 target sites were PCR amplified, cloned into pEASY-Blunt-Zero Cloning Kit (Beijing TransGen Biotech, Beijing, China) and then sequenced with M13 primer.

For detection of nucleus localized Cas9 proteins, 2 × 10^6^ of K562 cells were nucleofected with 100 μg Cas9 proteins and then harvested at indicated time points. Collected cells were fractionated using nuclear and cytoplasmic extraction kit (Beyotime Biotechnology, Shanghai, China) for western blot (WB) analysis. Cas9 proteins were detected using monoclonal anti-HA antibody (Cell Signaling Technology, Danvers, MA, USA, 2999) at 1:5000 dilutions. Nuclear internal control lamin B1 was detected using anti-Lamin B1 antibody (Cell Signaling Technology, 13435S) at 1:5000 dilutions.

### 2.6. Lipofection

HEK293-derived EGFP reporter cells (1 × 10^5^) were seeded on to a 24-well plate and transfected at 24 h after seeding. Cas9 proteins were diluted with 25 μL of Opti-MEM medium into indicated concentration, followed by addition of sgRNA and 1 μL Cas9 Plus (supplied in Lipofectamine CRISPRMAX kit, Life Technologies, Carlsbad, CA, USA). This solution was thoroughly mixed and then incubated at room temperature for 10 min. Thereafter, 1.5 μL of Lipofectamine CRISPRMAX transfection reagent (Life Technologies) was added to each tube. The mixture was incubated at room temperature for 15 min and then added to cell culture. The percentage of EGFP activation was measured by Cytoflex flow cytometer (Beckman, Brea, CA, USA) at 48 h post transfection.

### 2.7. Microinjection of Mouse Embryos

Microinjection was carried out at Shanghai Model Organisms Center Inc. (Shanghai, China) with the approval from Animal Care and Use Committee (2016-W-1516, 24 August 2016) using a described protocol [23]. Superovulated female C57BL/6J mice of 3–4-week-old were mated to male C57BL/6J mice of 8–12-week-old. Fertilized embryos were collected from oviducts. All experiments were conducted in accordance with the guidelines of the American Association for the Accreditation of Laboratory Animal Care (AAALAC). For Cas9 protein injection, 160 ng/μL of Cas9 proteins and 50 ng/μL sgRNA were co-injected into the pronucleus of fertilized eggs. All injected zygotes had well recognized pronuclei. After microinjection, zygotes were cultured in KSOM mouse embryo culture medium supplemented with amino acids at 37 °C under 5% CO_2_ until blastocyst stage at 3.5 days. For sequence analysis, CRISPR/Cas9 target sites were PCR amplified, cloned into pEASY-Blunt-Zero Cloning Kit (Beijing TransGen Biotech, Beijing, China) and then sequenced with M13 primer.

### 2.8. Western Blot

K562 cells (2 × 10^6^) were electroporated with 100 μg Cas9 proteins. Cells were harvested at indicated time points. The nuclear proteins were isolated from cell lysate by using nuclear and cytoplasmic extraction kit (Beyotime Biotechnology, Shanghai, China) and fractionated using a 4–12% Novex Bis-tris gel. The proteins were transferred to a PVDF membrane using an iBlot following the manufacturer’s protocol or using the typical wet-transfer method. Upon blocking, the PVDF membrane was incubated for overnight with monoclonal HA antibody (Cell signaling technology, 2999) at 1:5000 dilution, and the nuclear inner reference Lamin B1(Cell signaling technology, 13435S) at 1:5000 dilution. After washing, the membrane was incubated for 1 h with rabbit anti-mouse antibody-HRP conjugate at 1:5000 dilution. After extensive washing, the membrane was developed with Pierce ECL reagent, followed by imaging using a BioRad ChemiDoc imager.

## 3. Results

### 3.1. Production and Validation of Multi-NLS SpCas9 Proteins

As most previous studies have no more than 4 NLS repeats on Cas proteins, we constructed a series of Cas9 variants carrying 0 to 4 NLS repeats at N- or C-terminus of SpCas9, with a total number of NLS repeats up to 4 (Figure 2A). These Cas9 variants were constructed with a His_6_ tag for affinity purification and a hemagglutinin (HA) tag for western blotting (WB) analysis (Figure 2A). For clarity, the multi-NLS SpCas9 proteins are hereafter referred to as Na/Cb Cas9, where a and b indicate the number of NLS repeats at N- and C-termini respectively.

Following an established protocol [22], multi-NLS Cas9 proteins were purified to more than 90% homogeneity using one-step affinity purification with a few exceptions (Appendix A). The yield of each protein was in the range of 3 to 5 mg per liter culture. To assess the in vitro activity of multi-NLS SpCas9 variants, purified proteins were pre-assembled with in vitro transcribed *CCR5*-targeting sgRNA (Appendix A) and SpCas9 RNPs were added into the reaction containing the substrate DNA. Titration of SpCas9 RNPs showed that all SpCas9 variants could cleave substrate DNA at a RNP concentration of 5 nM or below (Appendix A). These results suggested that the impurities in the purified SpCas9 proteins did not compromise the in-vitro cleavage activity.

### 3.2. Evaluation of the Cellular Activity of Lipofected Multi-NLS SpCas9 Proteins in EGFP Reporter Cells

To facilitate the evaluation of the cellular activity of multi-NLS SpCas9, we employed a previously described HEK293-derived EGFP reporter cell line where the EGFP fluorescence could be activated by CRISPR-Cas9 targeting [22] (Figure 2B). Purified SpCas9 proteins in complex with EGFP-targeted sgRNA were packaged using a commercially available lipofection reagent CRISPRMAX and then transfected into the EGFP reporter cells. It was found that within the N0 subgroup, all SpCas9 variants carrying two or more NLS repeats at the C-terminus exhibited higher efficiency of EGFP activation than the single-NLS SpCas9 (N0/C1) (Figure 2C). By contrast, within the C0 subgroup, N2/C0, N3/C0 or N4/C0 SpCas9 did not show significantly higher activity than N1/C0 construct (Figure 2C). Importantly, N0/C4 construct appeared to display the highest rate of EGFP activation among the SpCas9 variants tested. These results indicated that C-terminal NLS was more important than N-terminal NLS for improving the cellular activity of purified SpCas9 proteins. Interestingly, N0/C0 construct without any fusion of NLS also exhibited genome-editing activity, suggesting that other components in SpCas9 RNP could contribute to nuclear localization.

### 3.3. Evaluation of the Cellular Activity of Nucleofected Multi-NLS SpCas9 Proteins

Next, we sought to analyze the activity of multi-NLS SpCas9 proteins at endogenous genomic loci. To investigate the general applicability of multi-NLS fusion, its effects under the context of different delivery approaches and cell types were analyzed. Pre-assembled RNPs harboring multi-NLS SpCas9 proteins were nucleofected into K562 cells and the genome-editing activity of different SpCas9 variants were quantified and validated by T7E1 and Sanger sequencing (Appendix A). Similar to the above results with lipofection, in nucleofected K562 cells addition of NLS at the C-terminus of SpCas9 significantly increased its cellular activity, by comparing N0/C2, N0/C3 or N0/C4 with N0/C1, while extra N-terminal NLS seemed to result in limited beneficial effects when N2/C0, N3/C0 or N4/C0 were compared with N1/C0 (Figure 3A). This result was consistent with the results with lipofection, suggesting that C-terminal NLS had more effects on the cellular activity of SpCas9. Interestingly, in the case of RNP nucleofection N1/C2 variant exhibited highest gene editing efficiency. Nevertheless, given the consistent performance of C-terminal multi-NLS fusion in lipofected and nucleofected SpCas9 proteins, we focused on analyzing C-terminal variants including N0/C1, N0/C2, N0/C3 and N0/C4 in the following experiments. Consistently, C-terminal multi-NLS improved the activity of nucleofected SpCas9 proteins in Jurkat cells with N0/C2 construct displaying highest rates of gene modification (Figure 3B and Appendix A).

Next, we sought to investigate the kinetics of multi-NLS SpCas9 proteins in cells. Western blotting analysis showed that the majority of N0/C1 SpCas9 protein was degraded after 2 h post nucleofection in K562 cells (Figure 3C). Comparison of nucleus-localized protein at 1 h post nucleofection revealed that C-terminal multi-NLS could enhance nuclear localization of SpCas9 proteins (Figure 3D) despite that the proteins were nucleofected into cells and were supposed to have high efficiency of nuclear localization [24].

### 3.4. Evaluation of the Genome-Editing Specificity of Multi-NLS SpCas9 Proteins Delivered via Nucleofection into K562 Cells

Next, we intended to investigate the effects of multi-NLS on the specificity of SpCas9 proteins. As the specificity of CRISPR-Cas9 is known to be dependent on the dosage of nucleases, sgRNA or their complexation ratio [25], we first analyzed the effects of multi-NLS on the on-target activity of N0/C1, N0/C2, N0/C4 and N1/C2 SpCas9 variants at different Cas9 and sgRNA dosages. It was found that at low dosage of SpCas9-sgRNA (15 μg: 3 μg), all multi-NLS displayed higher efficiency than N0/C1, with N0/C4 exhibiting highest genome-editing activity. When SpCas9 was fixed to 15 mg and sgRNA dosage was increased to 10 μg, multi-NLS Cas9 proteins N0/C2, N0/C4 and N1/C2 had similar activity, all surpassing N0/C1 protein. At high dosage of 30 μg SpCas9 and 30 μg sgRNA, N0/C2 and N0/C4 showed minor increase of editing efficiency in comparison with N0/C1, whereas N1/C2 construct displayed greatest improvement. Interestingly, at medium dosage of RNPs N1/C2 and N0/C4 had similar cellular activity (Figure 4A and Appendix A). These results suggested that the effects of multi-NLS on SpCas9 proteins were dependent on the dosage.

To determine the effects of multi-NLS fusion on the specificity of SpCas9 proteins, we chose to analyze the on-target activities of two previously described sgRNAs and their off-target activities at three genomic sites [26] under the context of multi-NLS SpCas9 variants. The on- and off-target activities were separately determined by T7E1 and the non-specific activity of SpCas9 was presented as off/on targeting ratio. It was found that compared with N0/C1, N1/C2 and N0/C4 SpCas9 proteins did not exhibit notably increased non-specificity, as indicated by the off/on ratio, for most conditions tested (Figure 4B and Appendix A). These results suggested that multi-NLS fusion could, by a large extent, retain the specificity of SpCas9 proteins.

### 3.5. Genome Editing of Primary Cells Using Nucleofection of Multi-NLS SpCas9 Proteins

To expand the applicability of multi-NLS fusion, we examined the genome-editing activity of multi-NLS SpCas9 proteins on therapeutically relevant human neural progenitor cells (hNPCs) and human dermal fibroblasts (hDFs). In hNPCs, addition of two or three extra NLS repeats to the C-terminus could result in nearly 2-fold higher editing activity than N0/C1. In contrast to the results in transformed cells, N0/C4 did not exhibit notable improvement. Interestingly, SpCas9 proteins without any NLS fusion (N0/C0) could also have 10% editing events (Figure 5A and Appendix A). The overall editing activity of SpCas9 proteins in hDFs are relatively lower compared with that in hNPCs. Nevertheless, N0/C3 exhibited greatest improvement in the editing activity (Figure 5B and Appendix A). In addition, we noticed that in neither hNPCs nor hDFs did N0/C4 exhibit highest editing efficiency. This could arise from the largely enhanced nuclear transportation of N0/C4 protein (Figure 3D), which might be toxic to the fragile progenitor and primary cells.

### 3.6. Genome Editing of Mouse Embryos Using Microinjection of Multi-NLS SpCas9 Proteins

To explore the effects of multi-NLS on SpCas9-mediated genome editing in mouse embryos, we chose to compare the activity of N0/C1 and N0/C4 variants. We first designed six sgRNAs targeting to murine *Pten* gene and screened for efficient sgRNAs for generating genomic modifications in mouse Hepa 1-6 cells (Figure 6A and Appendix A). sgRNAs 2 and 3 exhibited similar editing rates and surpassed other sgRNAs. sgRNA 2 was selected for the following microinjection experiment.

Pre-assembled N0/C1 and N0/C4 SpCas9-sgRNA was microinjected into the pronucleus of mouse zygotes. In total, 100 zygotes were injected for each SpCas9 variant and genomic DNA was successfully recovered from approximately 80 of 2-cell stage embryos for each SpCas9 construct. These embryos were pooled, and then the genome editing activity was determined and validated using T7E1 and Sanger sequencing. Both T7E1 and Sanger sequencing revealed nearly 2-fold higher activity of N0/C4 protein than N0/C1 construct (Figure 6B,C and Appendix A).

We next injected approximately 120 mouse zygotes with N0/C1 and N0/C4 SpCas9 RNPs and allowed the cells to develop to blastocyst stage. No major difference of cell development was observed between the embryos treated with N0/C1 and N0/C4 SpCas9 proteins (Figure 6D). In total, 21 and 19 blastocysts were obtained for N0/C1 and N0/C4 groups respectively. *PTEN* gene was successfully amplified from 14 and 9 blastocysts from the N0/C1 and N0/C4 groups, respectively. In these amplified *PTEN* sites, both single- and dual- allelic mutations were observed and N0/C4 appeared to have higher rates (7 out of 9) of genome editing than N0/C1 SpCas9 (5/14) (Figure 6E) despite the limited number of embryos analyzed.

## 4. Discussion

Multiple SV40 NLS fusion has been widely used in genome engineering tools for improving the targeting efficiency. This study focuses on a specific form of SpCas9 that is delivered as purified proteins. It has been well established that purified proteins can reduce the off-target activity of SpCas9 by limiting the exposure of host genome to nucleases [26]. In addition, Cas9 RNP does not pose the risk of genomic integration of nuclease sequence and thus is particularly useful for therapeutic applications. The present study systematically investigated the effects of N- and/or C-terminal fusion of multiple SV40 NLS on the genome editing activity of purified SpCas9 proteins.

The first interesting discovery was that N0/C0 variant harboring no NLS sequence displayed appreciable level of gene modification, in both the case of lipofection and nucleofection of SpCas9 RNPs. These results have suggested the ability of spontaneous nuclear localization of SpCas9 fusion protein and/or sgRNA. In future studies, it could be interesting to explore the sequence dependency of the nuclear localization of SpCas9 protein and sgRNA. On the other hand, by analyzing the kinetics of nucleofected SpCas9 proteins we provided experimental evidence that multi-NLS can indeed improve the nuclear localization efficiency of SpCas9.

A second discovery was that the C-terminal NLS fusion seemed to be more beneficial than N-terminal fusion for improving the genome editing activity of SpCas9 proteins. This is likely because fusion of excessive positively charged amino acids to the N-terminus may interfere with the active-site residue Asp10 in SpCas9 for DNA recognition and cleavage. The abilities to improve SpCas9 activity in primary cells and to retain the editing specificity render multi-NLS fusion a facile yet efficient strategy for expanding the therapeutic applications of SpCas9. Most importantly, the consistently improved cellular activity of SpCas9 proteins by multi-NLS fusion across different dosage, delivery approaches and cell types have suggested that nuclear localization of conventional SpCas9 design harboring single NLS is indeed a rate-limiting step. In addition, our results have shown that the exact effects of multi-NLS fusion may be context-dependent, relying on both the delivery approaches and cell types, as exemplified by our (Figure 2 and Figure 3) and previous results [21].

The most surprising result of multi-NLS fusion came from its ability to enhance the gene-editing efficiency of SpCas9 RNPs in embryos via microinjection. Although we controlled the microinjection process by directly injecting the Cas9 RNPs into the pronucleus of zygotes, markedly improved gene-editing efficiency was observed with N0/C4 Cas9 protein in comparison with N0/C1 Cas9 protein. It is likely that during the injection process, pronucleus-injected SpCas9 RNPs may diffuse to the cytoplasm of zygotes and multi-NLS could help SpCas9 to re-localize in the nuclei of zygotes. Given the advantages of SpCas9 RNPs in generating model animals, multi-NLS fusion thus provides a simple yet efficient approach to generating genetically modified organisms with high rates of success.

## 5. Conclusions

In the present study, we systematically investigated the effects of multi-NLS on the cellular activity of purified SpCas9 proteins and found that multi-NLS sequence can improve the genome-editing activity of SpCas9 across different delivery approaches and cell types without compromising the targeting specificity. Our study can help establish multi-NLS fusion—particularly C-terminal fusion—as an efficient approach to enhancing the cellular activity of SpCas9 in a cell type-dependent manner.

## Figures and Tables

**Figure 1 bioengineering-09-00083-f001:**
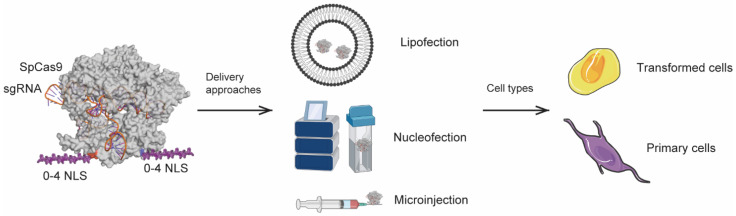
Schematic presentation of the experimental procedures in this study.

**Figure 2 bioengineering-09-00083-f002:**
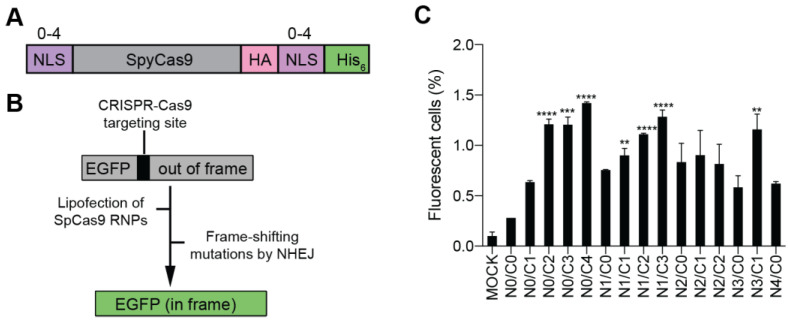
Evaluation of the cellular activity of multi-NLS SpCas9 proteins in HEK293-derived EGFP reporter cells. (**A**) Design of multi-NLS SpCas9 proteins. (**B**) Schematic representation of HEK293-derived EGFP reporter cells for evaluation of CRISPR-Cas9 cellular activity. (**C**) EGFP activation by lipofection of SpCas9 RNPs. SpCas9 proteins of 10 μg and sgRNA of 5 μg are complexed and transfected into reporter cells by CRISPRMAX. In total, 2 × 10^4^ cells are sorted. The data are shown as mean ± SD (*n* = 4). The multi-NLS SpCas9 variants with significantly improved EGFP activation in comparison with N0/C1 construct are indicated. Significant difference is calculated using Student’s *t* test (**, *p* < 0.01; ***, *p* < 0.001; ****, *p* < 0.0001). The difference between SpCas9 groups and mock group is determined using one-way analysis of variance (ANOVA) with Dunnett’s multiple comparisons test. Single- and multi-NLS groups have *P* values of less than 0.001 in comparison with mock group. N0/C0 is not significantly different from mock (*p* = 0.5020).

**Figure 3 bioengineering-09-00083-f003:**
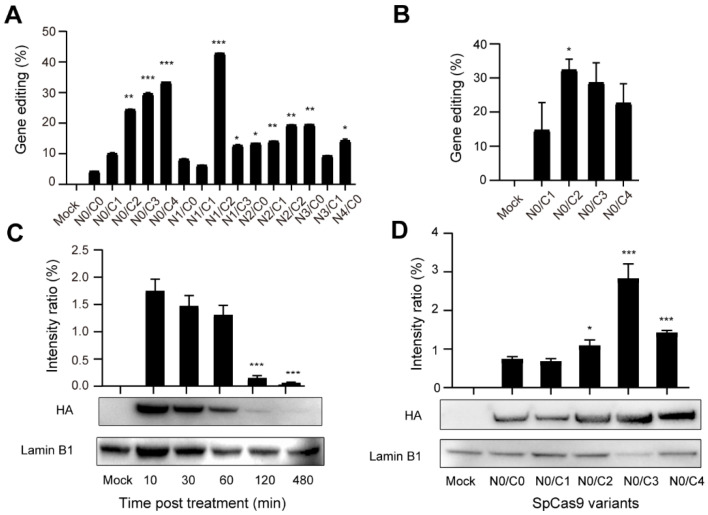
Evaluation of the genome-editing activity of nucleofected multi-NLS SpCas9 proteins. (**A**,**B**) Evaluation of the genome-editing activity of multi-NLS SpCas9 at the *CCR5* site in K562 (**A**) and Jurkat (**B**) cells. The results are quantified using T7E1 analysis and presented as mean ± SD (*n* = 2). The difference between N0/C1 and multi-NLS constructs is analyzed by two-tailed Student’s *t* test. *, *p* < 0.05; **, *p* < 0.01; ***, *p* < 0.001. The difference between SpCas9 groups and mock group is determined using one-way analysis of variance (ANOVA) with Dunnett’s multiple comparisons test. All SpCas9 groups have *P* values of less than 0.05 in comparison with mock group. (**C**) Analysis of the retention of nucleofected N0/C1 SpCas9 proteins in K562 cells. The difference between 10 min and other time points is analyzed by two-tailed Student’s *t* test. ***, *p* < 0.001. (**D**) Analysis of nucleus localized SpCas9 proteins at 1 h post nucleofection of K562 cells. The difference between N0/C1 and multi-NLS constructs is analyzed by two-tailed Student’s *t* test. *, *p* < 0.05; ***, *p* < 0.001.

**Figure 4 bioengineering-09-00083-f004:**
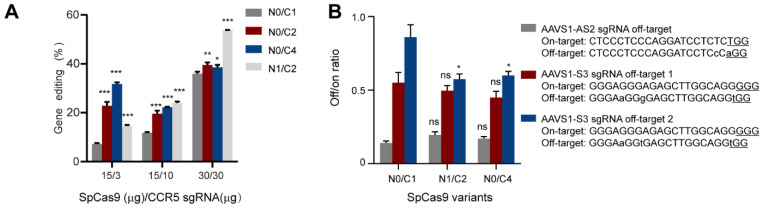
Evaluation of the effects of multi-NLS on SpCas9 activity and specificity in K562 cells. (**A**) The gene-editing activities of multi-NLS on SpCas9 at different dosage, as determined by T7E1 analysis. The data are shown as mean ± SD (*n* = 3 technical replicates). (**B**) The effects of multi-NLS on the specificity of SpCas9 variants. The specificity is determined by the ratio between on-target and off-target activities. The difference between N0/C1 and multi-NLS constructs at each corresponding condition is analyzed by one-way ANOVA with Bonferroni’s multiple comparisons test. *, *p* < 0.05; **, *p* < 0.01; ***, *p* < 0.001.

**Figure 5 bioengineering-09-00083-f005:**
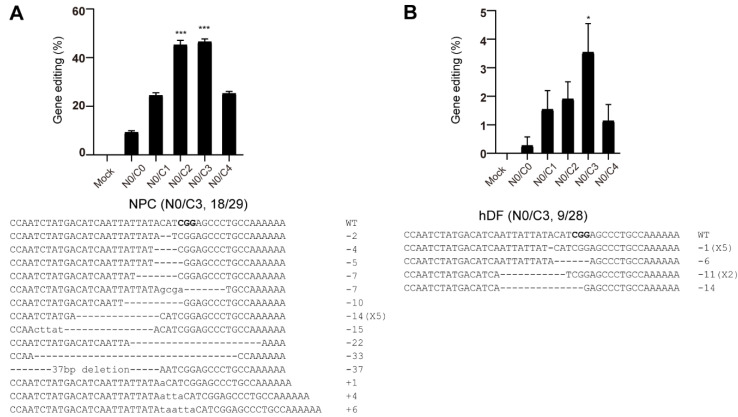
Genome editing of hNPCs (**A**) and hDFs (**B**) using multi-NLS SpCas9 proteins. The cells are nucleofected with 30 μg SpCas9 proteins and 30 μg *CCR5* sgRNA. The editing efficiencies are determined by T7E1 analysis and presented as mean ± SD (*n* = 3 technical replicates). Representative Sanger sequencing results are shown. PAM sequence is highlighted in bold. The Sanger-sequenced clones containing modified *CCR5* sites over the total sequenced clones are indicated as 18/29 and 9/28 respectively. The difference between N0/C1 and multi-NLS constructs is analyzed by two-tailed Student’s *t* test. *, *p* < 0.05; ***, *p* < 0.001. The difference between SpCas9 groups and mock group is determined using one-way analysis of variance (ANOVA) with Dunnett’s multiple comparisons test. In NPCs (**A**), all SpCas9 groups have *P* values of less than 0.001 in comparison with mock group. In Jukat cells (**B**), *p* = 0.975 for N0/C0, *p* < 0.05 for N0/C1 and N0/C2, *p* < 0.001 for N0/C3, *p* = 0.149 for N0/C4.

**Figure 6 bioengineering-09-00083-f006:**
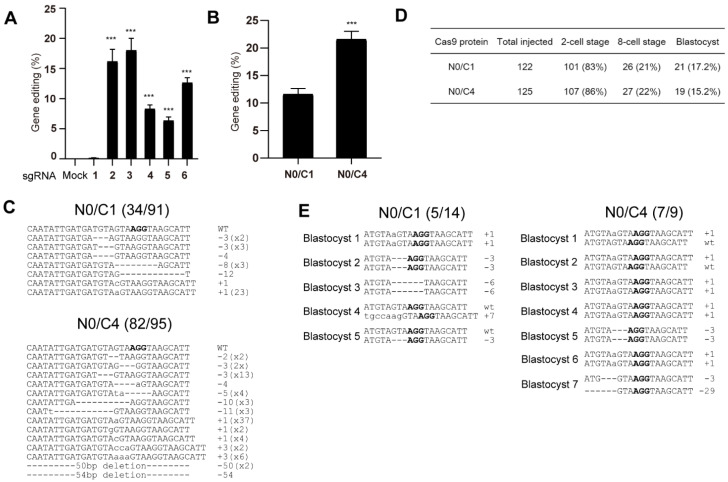
Genome editing of mouse embryos using microinjected N0/C1 and N0/C4 SpCas9 proteins. (**A**) Screening of PTEN-targeting sgRNA in mouse Hepa 1-6 cell line for genome editing. The editing efficiencies are determined by T7E1 analysis and presented as mean ± SD (*n* = 3 technical replicates). The difference between sgRNA-1 and other sgRNAs is analyzed by two-tailed Student’s *t* test. ***, *p* < 0.001. The difference between mock and sgRNAs is analyzed by one-way ANOVA with Dunnett’s multiple comparisons test. *p* < 0.001 for sgRNAs 2–6. (**B**) T7E1 analyses of pooled 2-cell embryos treated with N0/C1 and N0/C4 SpCas9 proteins respectively. The data are presented as mean ± SD (*n* = 3 technical replicates). The difference between N0/C1 and multi-NLS constructs is analyzed by two-tailed Student’s *t* test. ***, *p* < 0.001. (**C**) Sanger sequencing validation of mutated PTEN site in pooled embryos. (**D**) The effects of N0/C1 and N0/C4 SpCas9 proteins on the development of mouse embryos. (**E**) Sanger sequencing results of mutated PTEN sites in blastocysts. PAM sequence is highlighted in bold.

## Data Availability

Experimental data and materials can be provided upon reasonable request to J. Liu.

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
