# Peer review of "Systematic Investigation of the Effects of Multiple SV40 Nuclear Localization Signal Fusion on the Genome Editing Activity of Purified SpCas9"

_bioengineering, 2022, doi:10.3390/bioengineering9020083_

Round 1
Reviewer 1 Report
In this study, the authors used lipofection and nucleofection as well as different cell lines and mouse embryos to investigate the genome-editing activity of SpCas9 with different combinations of the nuclear localization signal (NLS). They found that SpCas9 with multi-NLS fusion at the C-terminal region was more efficient in genome-editing. As the authors stated, previous studies have already shown that multiple SV40 NLS fusion can improve the targeting activity of Cas9 in genome-editing, so the novelty of the present work is rather limited. In addition, the conclusion that multi-NLS fusion at the C-terminal region enhances genome-editing activity of SpCas9 is not convincing. I have several major points.
1. In lipofection experiments using HEK293 cells, N0C4 seems to display highest genome-editing activity. However, this is not the case in nucleofection experiments using K562 cells, in which N0C4 only shows a moderate activity (see figure 3B). Even in K562 cells, the results are not consistent. Figure 3A indicates that N1C2 displays a highest activity, while figure 3B shows that N0C2 may have a higher activity. It is curious that the activity of N1C2 was not assayed by T7E1 analysis.
2. It is unclear how the data in figure 3A were obtained and quantified.
3. It is also unclear how the authors determine genome-editing activity and specificity of SpCas9 by analysis of dose-response.
4. Another problem is that expression levels of SpCas9 in different conditions were not controlled, so it is unclear whether they are expressed similarly.
5. The activity of SpCas9 with multiple C-terminally fused NLS repeats has been tested by others. However, these works were not cited (for example, PMID: 31900423).
6. Figure 2C, it is unclear how many cells were counted.
Author Response
Comments and Suggestions for Authors
In this study, the authors used lipofection and nucleofection as well as different cell lines and mouse embryos to investigate the genome-editing activity of SpCas9 with different combinations of the nuclear localization signal (NLS). They found that SpCas9 with multi-NLS fusion at the C-terminal region was more efficient in genome-editing. As the authors stated, previous studies have already shown that multiple SV40 NLS fusion can improve the targeting activity of Cas9 in genome-editing, so the novelty of the present work is rather limited. In addition, the conclusion that multi-NLS fusion at the C-terminal region enhances genome-editing activity of SpCas9 is not convincing. I have several major points.
- In lipofection experiments using HEK293 cells, N0C4 seems to display highest genome-editing activity. However, this is not the case in nucleofection experiments using K562 cells, in which N0C4 only shows a moderate activity (see figure 3B). Even in K562 cells, the results are not consistent. Figure 3A indicates that N1C2 displays a highest activity, while figure 3B shows that N0C2 may have a higher activity. It is curious that the activity of N1C2 was not assayed by T7E1 analysis.
Response: We apologize for the confusion. Figure 3B was in fact an experiment performed in Jurkat cells. The effects of NLS on SpCas9 editing might be different in different cell lines. We have revised the text to clearly indicate this. In addition, we did observe that the effects of NLS of SpCas9 editing were dependent on the delivery approaches, as the reviewer noticed. We have revised the discussion to provide additional explanation for this observation.
- It is unclear how the data in figure 3A were obtained and quantified.
Response: The results were obtained from T7E1 assay and quantified as described in the Materials and Methods section. We have revised the Figure caption to clearly indicate this.
- It is also unclear how the authors determine genome-editing activity and specificity of SpCas9 by analysis of dose-response.
Response: The results were obtained from T7E1 assay as mentioned above. The specificity was determined as the ratio between on-target and off-target activities as previously described (Gaj et al., 2012, Nat Methods). We have revised the Figure caption to provide these details.
- Another problem is that expression levels of SpCas9 in different conditions were not controlled, so it is unclear whether they are expressed similarly.
Response: We thank the reviewer for raising this question. The SpCas9 proteins were all expressed in E. coli cells and purified using the same conditions. The yield of each protein sample was also similar, generally in the range of 3-5 mg per liter culture. We have revised the Materials and Methods and Results sections to provide more detailed description.
- The activity of SpCas9 with multiple C-terminally fused NLS repeats has been tested by others. However, these works were not cited (for example, PMID: 31900423).
Response: We apologize for missing this important literature and have revised the Introduction and Discussion sections to include this reference along with additional discussion.
- Figure 2C, it is unclear how many cells were counted.
Response: We have revised the figure caption to include this information.
Reviewer 2 Report
In the present article Shui et al. generated and tested several Cas9 variants by including multiple SV40 NLS either at C and N terminal of the protein. Demonstrating that in some specific cases, the newly generated proteins have superior performance in editing but without getting a higher risk of off-target effects.
Major points.
In my opinion, the reporter system used at the beginning of the manuscript (eGFP out of frame reporter) is not the best option because at the end it just accounts for the 1/3 of cases in which the indel repairs are in the same frame and some information is lost in the 2/3 of the cases that maybe there are others editing outcomes. Maybe, sequencing the results either by NGS or by Sanger (through the comparison with a control chromatogram, TIDE analysis) would give the authors a deeper understanding of what is happening and the real % of editing. In the same way, T7EI essay is not a standard to measure gene editing efficiencies as it is unable to detect in an accurate way 1 base indels, so it would be better to analyze by means of sequence trace decomposition (TIDE or ICE analysis).
Regarding figure 3 (page 6) I do not understand why the authors use K562 cells in panels A, C, and D, and Jurkat just in panel B. Maybe using both cell types in the whole set of experiments would give them a better understanding of which variant works better and it could bypass cell specificity (note that it behaves in a different manner between panels A and B). keeping the focus on the same figure, I miss statistical analysis for panel A, it would be necessary to add it in order to have statistical power and not just the result from 1 experiment.
The authors mentioned that “as the specificity of CRISPR-Cas9 is known to be dependent on the dosage of nuclease” (line 245), this is a partial assumption because in fact it has been demonstrated that it is a balance between nuclease and sgRNA and in some scenarios sgRNA is even more important than Cas9; even more, the same authors demonstrated it on figure 4 where they use 15ug of Cas9 either with 3 or 10 ugs of sgRNA and in 5 out of 6 they have higher efficiency with the same amount of Cas9; so this sentence should be modified accordingly. In this same context, it is difficult to ascertain a clear result, because Cas9 and sgRNAs vary in concentration in the whole experimental design. I think that this experiment should be repeated just by modifying 1 of the variables (Cas9) in order to obtain a clear result that could lead to an unequivocal assumption.
Finally, the authors tested some of the Cas9 variants (I clearly miss N1C2 variant) in primary cells (hNPCs and hDFs) and oocytes. It should be explained in more detail why they are getting lower editing efficiencies with N0/C4 variant and demonstrate what’s the reason for obtaining it (maybe toxicity?).
Minor points.
Figure 2 legend (line 205) has two odd symbols that are supposed to be micrograms. Please modify it accordingly.
Round 2
Reviewer 1 Report
The issues raised in my previous review were not addressed in a satisfactory manner in this revised version. Authors’ responses and data presented in the manuscript raise additional concerns regarding the novelty and the conclusions of this work.
- Previous studies by other groups have already shown that C-terminal fusion of SpCas9 with multiple NLS repeats increases genome editing efficiency. Genome editing using purified Cas9 has been also reported. Thus the novelty of this work is not clear.
- It is confusing that the effects of NLS on SpCas9 editing strongly vary between cell lines. It seems that two NLS repeats at the C-terminus are efficient in different cell lines, and four repeats are not efficient in some cell lines. Thus, it is not appropriate to conclude that C-terminal NLS repeats (N0/C4) exhibited superior performance.
- The analysis of SpCas9 specificity is problematic. The authors state that they used three previously reported sgRNAs, but the designation of the three off-target sgRNAs is not exactly the same as in the literature, so it is not possible to find the corresponding sequences. There is also no information on the sequence of on-target sgRNAs. In addition, the authors only presented a graph with on/off ratio, but without showing the images of T7E1 assays. By the way, I cannot find supplementary figures and table S2 for primers in the submission system.
- It is not clear assays shown in figure 4 were performed in which cell line.
- It is disappointing that the analyses of genome editing efficiency using T7E1 assays were sometimes presented as graphs, sometimes as images. By the way, the “m” in different figures should be explained in the legends.
Round 3
Reviewer 1 Report
At the end of Introduction, the authors should briefly present the novelty of the study and their findings.
For many figures (3A,B, 4A,B, 5A,B and 6A,B), no statistical significance was indicated.
